# Unraveling the power of NAP-CNB's machine learning-enhanced tumor neoantigen prediction

**Almudena Mendez-Perez[1], Andres M Acosta-Moreno[1], Carlos Wert-Carvajal[1,2], Pilar Ballesteros-Cuartero[1], Ruben Sánchez-García[1,3], Jose R Macias[1], Rebeca Sanz-Pamplona[4,5], Ramon Alemany[6], Carlos Oscar Sorzano[1], Arrate Munoz-Barrutia[2], Esteban Veiga[1]\***

[1]Centro Nacional de Biotecnología, Consejo Superior de Investigaciones Científicas, Madrid, Spain; [2]Departamento de Bioingenieria, Universidad Carlos III de Madrid, Leganés, Madrid, Spain; [3]University of Oxford, Department of Statistics & XChem, Oxford, United Kingdom; [4]Catalan Institute of Oncology (ICO), Oncobell Program, Bellvitge Biomedical Research Institute (IDIBELL), L'Hospitalet de Llobregat, Barcelona, Spain; [5]University Hospital Lozano Blesa, Aragon Health Research Institute (IISA), ARAID Foundation, Aragon Government, Zaragoza, Spain; [6]Procure Program, Institut Català d'Oncologia-Oncobell Program, Catalan Institute of Oncology (ICO), Oncobell Program, Bellvitge Biomedical Research Institute (IDIBELL), L'Hospitalet de Llobregat, Barcelona, Spain

**\*For correspondence:**
eveiga@cnb.csic.es

**Competing interest:** The authors declare that no competing interests exist.

## eLife Assessment

Veiga et al demonstrate the importance of incorporating RNAseq and machine learning approaches for neoantigen prediction. The evidence is **convincing**, and these findings contribute **important** information towards the selection of neoantigens for personalized antitumor vaccination.

**Abstract** In this study, we present a proof-of-concept classical vaccination experiment that validates the in silico identification of tumor neoantigens (TNAs) using a machine learning-based platform called NAP-CNB. Unlike other TNA predictors, NAP-CNB leverages RNA-seq data to consider the relative expression of neoantigens in tumors. Our experiments show the efficacy of NAP-CNB. Predicted TNAs elicited potent antitumor responses in mice following classical vaccination protocols. Notably, optimal antitumor activity was observed when targeting the antigen with higher expression in the tumor, which was not the most immunogenic. Additionally, the vaccination combining different neoantigens resulted in vastly improved responses compared to each one individually, showing the worth of multiantigen-based approaches. These findings validate NAP-CNB as an innovative TNA identification platform and make a substantial contribution to advancing the next generation of personalized immunotherapies.

## Introduction

A new window of hope to treat previously intractable tumors is emerging through immunotherapies (*Chen and Mellman, 2013*). However, the response rates of these therapies remain low and relapses are common (*Kalbasi and Ribas, 2020*; *Novello et al., 2023*). Moreover, the severe undesired side

effects induce many patients to abandon the treatments (*Kalbasi and Ribas, 2020*), highlighting the urgent need for more specific novel therapies.

In this regard, the main challenges for most anticancer immunotherapies are the identification of tumor-specific antigens (neoantigens) (*Schumacher and Schreiber, 2015*) to avoid undesired side effects and the development of multiantigen-based treatments with combined therapies to prevent tumor relapses (*Cable et al., 2021*). A possible strategy to accelerate the search for neoantigens and lower the cost of the therapy is to sequence the cancer cells using next-generation sequencing techniques and find mutations using bioinformatics tools. Finally, it tries to predict which of the mutations will be more likely to cause an immune response, i.e., neoantigen prediction. This area is relatively unexplored with only a few algorithms available (*Boegel et al., 2019*). The need to validate the algorithmic results has already been recognized as one of the critical steps of this approach (*Vitiello and Zanetti, 2017*) and this work specifically addresses it. Algorithmic proposals using deep learning have only started to appear and most of them clearly outperform the standard approaches (*Bulik-Sullivan et al., 2018*; *Wu et al., 2019*). Finding neoantigens in every cancer patient will be fundamental for personalized antitumor immunotherapies (*Tran et al., 2017*).

We previously developed an easy-to-use platform (NAP-CNB) (NAP stands for neoantigen prediction) that allows to identify tumor neoantigens rapidly (*Wert-Carvajal et al., 2021*). NAP-CNB predicts putative neoantigens employing exclusively RNA-seq reads (*Wert-Carvajal et al., 2021*). The tool uses a long short-term memory-based neural network to rank mutations according to their estimated major histocompatibility complex (MHC) I affinity. NAP-CNB harnesses the suitability of recurrent neural networks for sequential problems to offer high accuracy in affinity binding prediction. Hence, NAP-CNB provides an integrated and resource-efficient pipeline for in silico classification of MHC I neoepitopes. In contrast with other tools (*Bjerregaard et al., 2019*; *Duan et al., 2014*; *Hasegawa et al., 2020*; *Hundal et al., 2016*), NAP-CNB is entirely automatic and freely available online.

We found NAP-CNB to be comparable or superior to other state-of-the-art methods of murine immunogenicity prediction, like NetH2pan (*DeVette et al., 2018*) or MHCflurry 2.0 (*O'Donnell et al., 2020*), in blind benchmarking. NAP-CNB presents an area under the curve of 95% for H-2K$^b$ typings and a high positive predictive value. The results improved with post-processing which consists of a majority voting method of an ensemble of sequences presenting single amino acid substitutions. Post-processing offers a more robust scoring by substituting each amino acid for its most similar one and then classifying the ensemble as the most repeated class.

## Results and discussion

In this work, we analyze in vivo the tumor neoantigen (TNA) predictive capabilities of NAP-CNB platform using the well-known murine B16 F10 melanoma as a model. For that, we synthesized the peptides corresponding to predicted TNA, used them to vaccinate mice, and analyzed the effectivity against tumor development.

In silico analysis of the B16 F10 melanoma gene expression showed several putative TNA (*Wert-Carvajal et al., 2021*) with different scores (predicted probability to be a TNA) and distinct gene expression quantified as fragments per kilobase million. We chose for peptide synthesis three top-scored TNA peptides (*Figure 1A*); *Pnp (low expression), *Adar (very low expression), *Lrrc28 (low expression). The * indicates that they are the mutated version, as indicated in the figure. The bottom-scored peptide, *Herc6 (high expression), therefore, predicted to not induce any immune reaction against the tumor, was chosen as a negative control. In addition, and in order to test whether the post-processing process offers some advantage, a top-scored peptide, *Wiz (high expression), revealed after post-processing was also synthesized (*Figure 1A*).

### Immunogenicity analysis

To analyze the immunogenicity of the predicted TNAs, we employed immunocompetent C57BL/6J mice as the recipient model. These mice were vaccinated with individual synthetic peptides *Pnp, *Adar, *Lrrc28, *Wiz, and *Herc6, the latter serving as a putative negative control. In all cases, peptides were emulsified in Freund's adjuvant. A second vaccination boost was injected 2 weeks after the first inoculation. Four weeks after the first inoculation the immunogenicity of the predicted antigens was

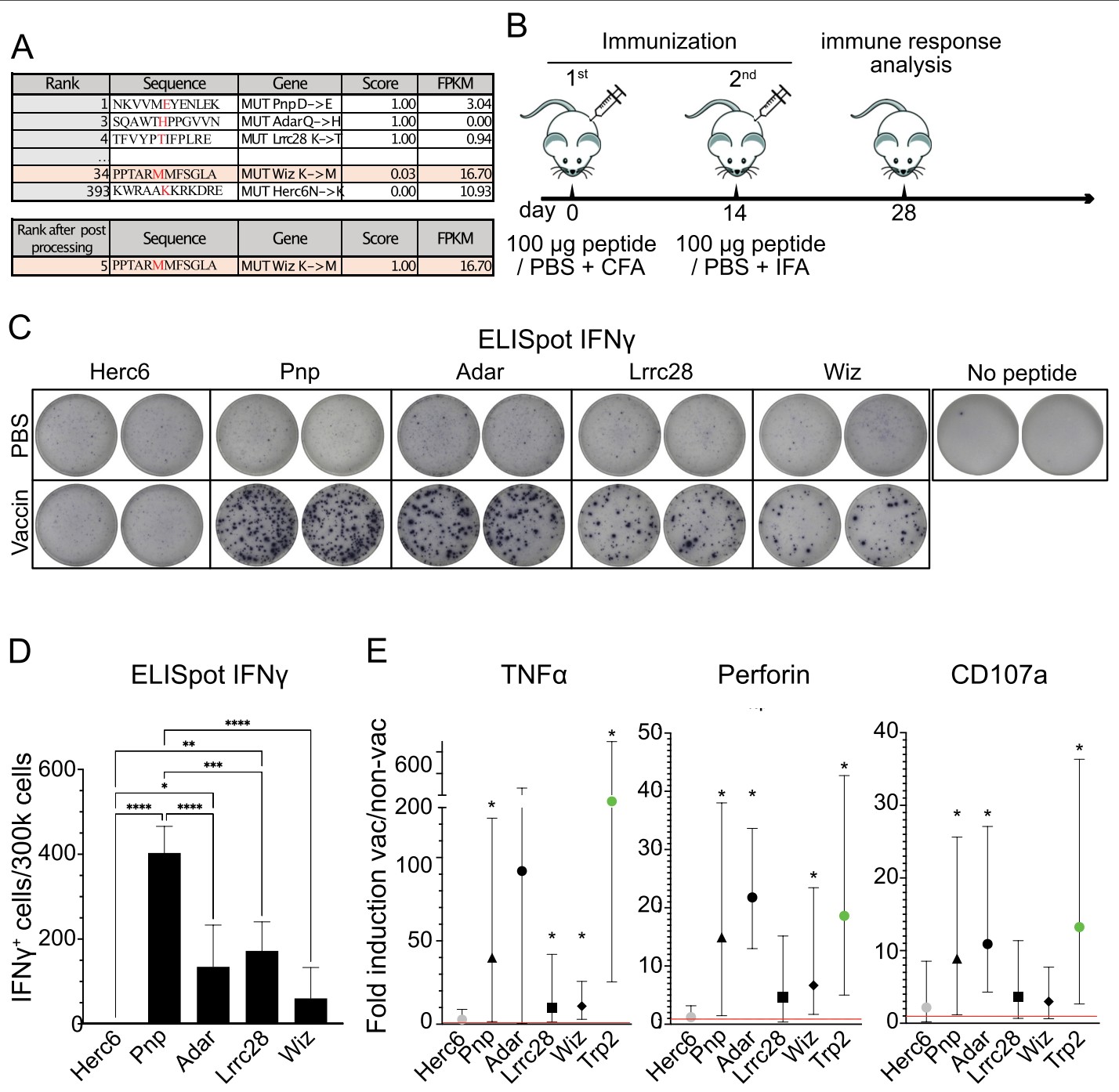

**Figure 1.** Vaccination-induced immune responses. (**A**) Putative B16 tumor neoantigen (TNA) identified by using the NAP-CNB platform ranked by scores of peptide sequences for a complete 12mer sequence. The TNA sequence, the mutation exclusive to tumor cells (in red), and gene name are shown. The gene expression is quantified as fragments per kilobase million (FPKM). The TNA score is also indicated. (**B**) Scheme of immunization. Two doses of peptides emulsified in Freund's adjuvant were subcutaneously (s.c.) injected, separated by 14 days. 14 days after the last dose the efficacy of the vaccine was analyzed by enzyme-linked immunosorbent spot (ELISpot) and intracellular cytokine staining (ICS) assays. (**C**) ELISpot analysis of interferon-gamma (IFNγ)-producing T-cell effectors from mice vaccinated with the indicated mutated peptides. The upper images show the response of non-vaccinated (phosphate-buffered saline [PBS]) animals after stimulation with the indicated peptides, the bottom images show the response of the animals vaccinated (vaccin) with the indicated peptides after restimulation with the same peptides. It is shown duplicates from representative animals. (**D**) As in C but showing the mean and sd of 5 mice/group. Asterisks indicate statistically significant differences analyzed by one-way ANOVA (* represent $p<0.05$, **$p<0.005$, ***$p<0.0005$). (**E**) ICS analysis of CD8[+] T-cells expressing tumor necrosis factor alpha (TNFα), perforin, or CD107a from mice vaccinated with the indicated mutated peptides: *Pnp (black triangle), *Adar (black circle), *Lrrc28 (black square), *Wiz (black rhombus), *Herc6 (gray

*Figure 1 continued on next page*

**Figure 1 continued**

circle), and Trp2 (green circle). The mean of the ratio of vaccinated divided by unvaccinated is shown, as the 95% confidence intervals. Intervals that do not include 1 and are therefore statistically significant are marked with *. 5 mice/group were used.

The online version of this article includes the following source data for figure 1:

**Source data 1.** Original data of *Figure 1C-E*.

tested (*Figure 1B*). A group of mice were treated with phosphate-buffered saline (PBS)+adjuvant (non-vaccinated control).

We assessed specific cellular responses against the predicted TNAs using an enzyme-linked immunosorbent spot (ELISpot) assay (*Figure 1C and D*). Compared to the control group immunized with *Herc6, all vaccinated groups displayed notably elevated levels of interferon-gamma (IFNγ)-secreting cells targeting TNAs. Notably, *Wiz vaccination yielded a comparatively weaker immune response. This experiment further validates the predicted non-immunogenic nature of the *Herc6 mutation.

Furthermore, we evaluated the immunogenic potential of the predicted TNAs through intracellular cytokine staining (ICS) (*Figure 1E*). The CD8+ T-cell population from vaccinated mice was activated by presenting the TNAs on splenocytes, and the production of tumor necrosis factor alpha (TNFα), perforin, and CD107a was analyzed via flow cytometry. Vaccination with all TNAs significantly induced the expression of at least one of the specified proteins after specific stimulation. Additionally, we included a well-established positive control, tyrosinase-related protein-2 (Trp2), a well-known tumor-associated antigen that induces potent cellular responses (*Castle et al., 2012*). As predicted, vaccination with *Herc6 did not elicit immunogenic responses.

These data show that silico-predicted TNAs by NAP-CNB induce robust immune responses following immunization in mice.

## In vivo antitumor implantation assays

The antitumor immunogenicity of predicted TNAs was further assessed using immunocompetent C57BL/6J mice, vaccinated with each synthetic peptide or with a peptide mixture containing *Pnp, *Adar, and *Lrrc28, in all cases emulsified in Freund's adjuvant. A second vaccination boost was injected 2 weeks after the first inoculation and 2 weeks before tumor implantation (B16 F10 melanoma; see experimental scheme in *Figure 2A*). Tumor size, as well as overall survival, was monitored over time.

The implanted tumor exhibited robust growth in the control group of non-vaccinated animals (PBS; *Figure 2B*). 15 days after implantation, all animals in this group displayed tumors with a volume exceeding 100 mm³. All animals died 22 days after tumor implantation (*Figure 3A and B*). Vaccination with the positive control, Trp2, showed immune responses against B16 F10 detected as slower tumor growth (*Figure 2C*) and incremented survival rates (*Figure 3A*). The median of the tumor growth in control unvaccinated mice (PBS) and vaccinated with Trp2 (positive control) are shown to compare the antitumor response of animals vaccinated with the indicated peptides. As expected, the immunization with *Herc6, in silico predicted as non-immunogenic, did not result in any positive outcome either in tumor growth (*Figure 2D*) or in survival rate (*Figure 3A*). Vaccination with *Pnp and *Lrrc28 elicited irregular antitumor responses (*Figure 2E and G*) with some mice showing excellent responses and other behaving similar to the negative control. These vaccinations showed slightly (not significant) incremented survival rates compared to control (*Figure 3B*). Vaccination with *Adar (the less expressed TNA; *Figure 1A*) did not reduce the tumor growth (*Figure 2F*) nor show any survival improvement (*Figure 3B*). However, the vaccination with a mixture of the three predicted TNA (Pnp, Adar, Lrrc28) induced a strong antitumor response, comparable and even better than that observed with the positive control, Trp2 (*Figure 2H*), resulting in a significantly increased survival (*Figure 3A*). In the same regard, the response against tumors in animals vaccinated with *Wiz (predicted TNA after post-processing and highly expressed) alone also elicited a powerful antitumor response (*Figure 2I*) with considerably improved survival (*Figure 3A*).

Additionally, as a measure of the vaccination effectiveness, it is shown the percentage of mice with tumors remained smaller than 100 mm³, 15 days after implantation (*Figure 2J*). Non-vaccinated animals or animals vaccinated with *Herc6 or *Adar show unsuccessful therapies. Vaccination with *Pnp or *Lrrc28 presented a successful rate of 75% and 50% respectively, and the maximum effectiveness,

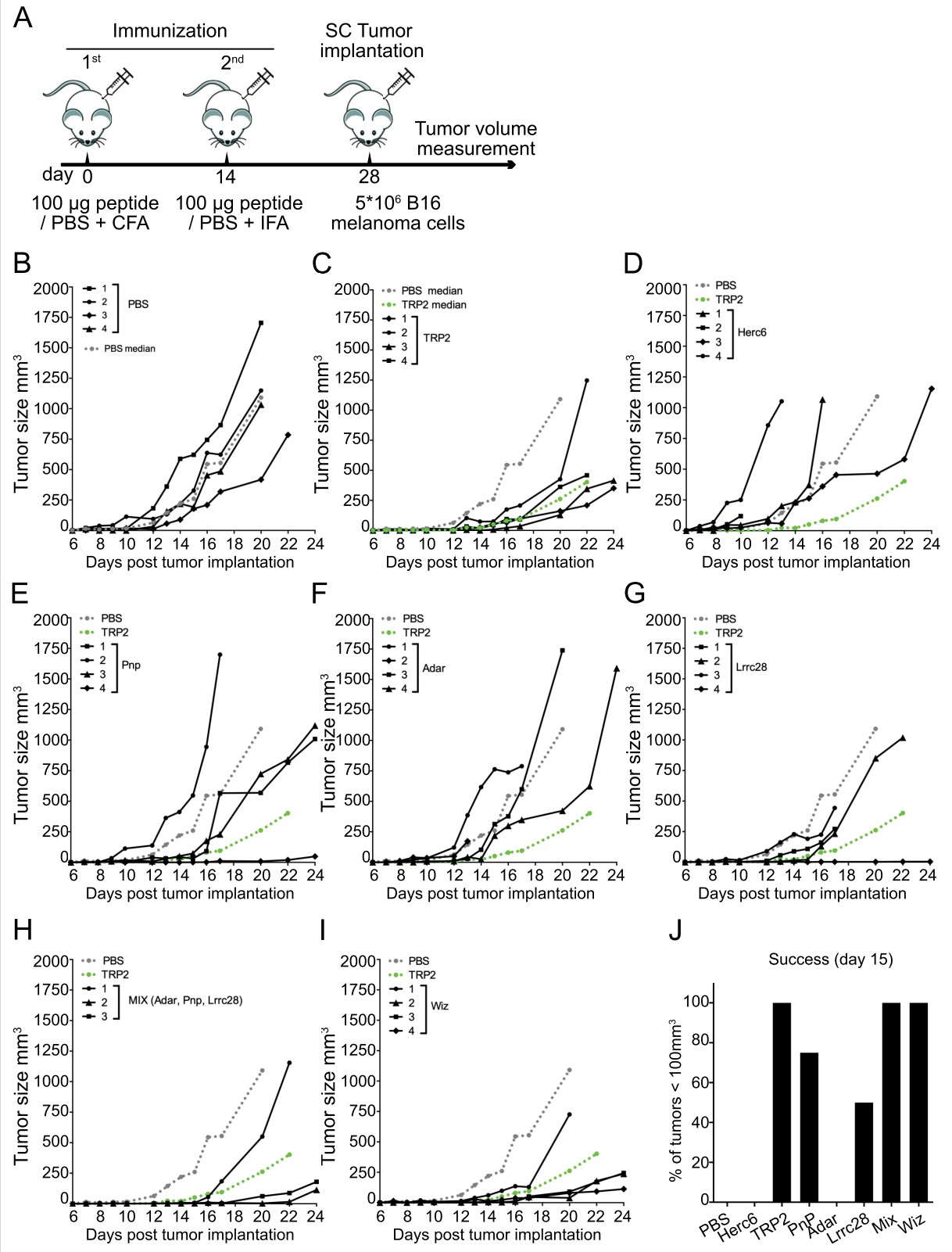

**Figure 2.** Antitumor activity of vaccination with predicted tumor neoantigen (TNA). (**A**) Scheme of immunization. Two doses of peptides emulsified in Freund's adjuvant were subcutaneously (s.c.) injected, separated by 14 days. 14 days after the last dose the B16 F10 melanoma cells were injected s.c. in the mid-right flank of C57BL/6J host mice and the tumor size over time was analyzed using a dial caliper. (**B–I**) Tumor growth on non-vaccinated (B) or vaccinated mice with the indicated peptide (**C–I**) (or peptide mix (**H**)), monitored every 1–3 days. Each line corresponds to the tumor size in one animal.

*Figure 2 continued on next page*

*Figure 2 continued*

The median of the tumors in non-vaccinated (phosphate-buffered saline [PBS]) and vaccinated with Trp2 (positive control) are shown as dashed lines in gray and green color, respectively. Mice with tumors≥900 mm³ were sacrificed. Following the rules of our ethical committee, animals presenting ulcers were also sacrificed. (**J**) Percentage of animals showing tumors with a volume≤100 mm³ 15 days after implantation. 4 mice/group were used.

The online version of this article includes the following source data for figure 2:

**Source data 1.** Original data of *Figure 2B-J*.

100% of animals, was observed in animals vaccinated with Trp2, *Wiz, or a peptide mixture (*Pnp, *Adar, and *Lrrc28).

Together, these data show that pure in silico approaches based on machine learning algorithms are able to identify TNAs that induce strong antitumor protection, which is the major bottleneck for most immunotherapies. The rapid identification of tumor neoantigens would allow to target/attack tumors non-treatable today and will vastly improve current immunotherapies, representing a giant step forward in the global anticancer fight. In this context, we demonstrate that the algorithms running in NAP-CNB platform effectively identify TNA that can be used as anticancer targets. The proof-of-concept experiments presented herein significantly bolster the prospects of translating TNA identification into practical applications for personalized cancer treatments within society.

Our findings also offer valuable insights for shaping future antitumor interventions. From a pragmatic standpoint, prioritizing TNAs with higher expression levels in the tumor, even if they elicit a comparatively weaker immunogenic response, appears to be a more promising approach. Our data confirm that employing a multiantigen therapy, targeting various tumor epitopes simultaneously, holds significant potential in averting immune escape. This underscores the importance of advancing TNA-based immunotherapy treatments against cancer within the framework of personalized medicine.

## Materials and methods
### Peptides
*Pnp (SLITNKVVMEYENLEKANHM), *Adar (LVPLSQAWTHPPGVVNPDSC), *Lrrc28 (EPMFTFVYPTIF-PLRETPMA), *Herc6 (SLVKKWRAAKKRKDREGAKR), *Wiz (TASPPPTARMMFSGLATPSL), and *Trp2 (PQIANCSVYDFFVWLHYYSV) were used for mice immunization. These peptides were synthesized at the proteomics unit from CNB-CSIC. The peptides were synthesized using the stepwise solid-phase

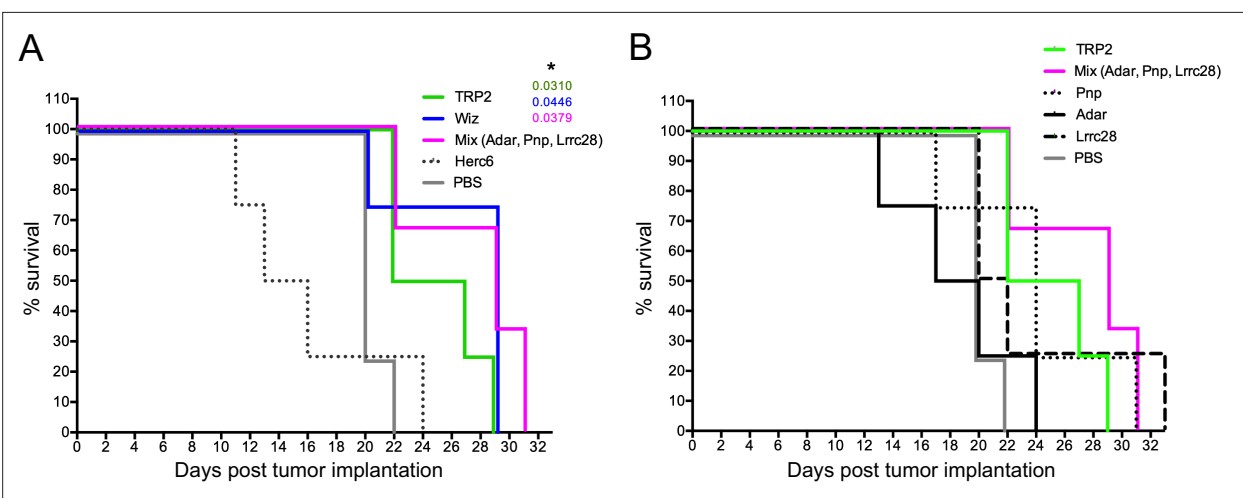

**Figure 3.** Survival curves of vaccinated mice. (**A, B**) Kaplan-Meier survival curves of mice vaccinated with the indicated tumor neoantigen (TNA) peptides and challenged with B16 F10 melanoma. Comparison of survival curves has been performed using the log-rank (Mantel-Cox) test. The significant p values comparing the control (phosphate-buffered saline [PBS]) group with each other group are indicated and * represents p<0.05. 4 mice/group were used.

The online version of this article includes the following source data for figure 3:

**Source data 1.** Original data of *Figure 3*.

peptide synthesis method performed on an automated peptide synthesizer (Multipep, Intavis, Köln, Germany). The amino acid polymerization was carried out using the standard Fmoc (*N*-(9-fluorenyl) methoxycarbonyl) chemistry and PYBOP/*N*-methylmorpholine as coupling activation reagents. The Fmoc-derivatized amino acid monomers and the preloaded resins used as support were obtained from Merck. Once synthesized the peptides were cleaved from the resin with a standard scavenger-containing trifluoroacetic acid (TFA)-water cleavage solution and precipitated by addition to cold ether. The crude peptides were purified by reverse-phase chromatography on a semi-preparative HPLC system (Jasco, Tokio, Japan) equipped with a C18 reversed-phase column (Scharlab, Barcelona, Spain). A linear gradient from 5% to 60% solvent B (0.05% TFA in 95% acetonitrile) in solvent A (0.05% TFA in water) was applied for 20 min. The chosen fractions were analyzed in a MALDI TOF 4800 mass spectrometer (Applied Biosystems, Framingham, MA, USA) and those containing the peptide were lyophilized. The peptides were then reconstituted to a concentration of 1 mg/ml in sterile PBS and preserved at –80°C.

## Mouse immunization

Equal volumes of immunogens and Freund's adjuvant (Imject Freund's Complete Adjuvant (FCA) for the first dose and Imject Freund's Incomplete Adjuvant (FIA) for the second dose, from Thermo Scientific) were mixed with a double-hub needle until a thick emulsion developed. The final immunogen concentration in the mixtures was 50 μg/100 μl. C57BL/6J mice were divided into eight groups and were subcutaneously (s.c.) immunized in the left flank with PBS (group 1) or with 100 μl of the emulsions containing *Trp2 peptide (group 2), *Herc6 peptide (group 3), *Pnp peptide (group 4), Adar peptide (group 5), *Lrrc28 peptide (group 6), *Pnp, *Adar, and *Lrrc28 peptides mixture (group 7), or *Wiz peptide (group 8). Immunizations were performed at day 0 (with FCA) and day 14 (with FIA). At day 28, the efficacy of the vaccine was analyzed by ELISpot and ICS assays, and an antitumor experiment was performed.

## ELISpot assay

The ELISpot assay was used to detect *Pnp, *Adar, *Lrrc28, *Herc6, and *Wiz specific IFNγ-secreting cells. 96-well nitrocellulose-bottom plates pre-coated with anti-mouse IFNγ monoclonal antibody were purchased from Mabtech. The plates were blocked with RPMI-10% fetal bovine serum (FBS) for at least 30 min. After spleen processing, $3 \times 10^5$ splenocytes per condition were restimulated with 1 μg/ml of the corresponding peptide pool or with RPMI-10% FBS. The plates were incubated with the peptides for 48 hr at 37°C in 5% $CO_2$ atmosphere, washed five times with PBS, and incubated with 1 μg/ml of the biotinylated anti-mouse IFNγ monoclonal antibody R4-6A2 (Mabtech) diluted in PBS-0.5% FBS for 2 hr at room temperature. The plates were then washed five times with PBS and a 1:1000 dilution of alkaline phosphatase (ALP)-conjugated streptavidin (Mabtech) was added. After 1 hr at room temperature, it was washed five times with PBS, and finally developed by adding the ready-to-use substrate solution BCIP/NBT-plus 5-bromo-4-chloro-3-indolyl phosphate/nitro blue tetrazolium chloride (Mabtech). The reaction was stopped by washing the plate with abundant water. Once it was dry, the spots were counted using the ELISpot Reader System - ELR02 - plate reader 651 (AID Autoimmun Diagnostika GmbH) with the aid of AID ELISpot reader system software (Vitro).

## ICS assay

The different CD8+ T-cell adaptive immune responses were analyzed by ICS as follows. After spleen processing, $4 \times 10^6$ fresh splenocytes (depleted of red blood cells) were seeded on M96 plates and stimulated for 12 hr in complete RPMI 1640 medium supplemented with 10% FBS containing 1 μl/ml Brefeldin A (BD Biosciences) to inhibit cytokine secretion, 1 μl/ml monensin 1× (eBioscience), anti-CD107a-FITC (BD Biosciences), and the peptides *Pnp, *AdaR, *Lrrc28, *Herc6, *Wiz, or Trp2 (1 μg/ml). Cells were then washed, stained for surface markers, fixed (eBioscience IC Fixation Buffer), permeabilized (eBioscience Permeabilization Buffer), and stained intracellularly with the appropriate fluorochromes. The fluorochrome-conjugated antibodies used for functional analyses were CD3-brilliant violet (BV)-510 (BD Biosciences), CD8 allophycocyanin (APC)-EFluor780 (eBioscience), PERFORIN-APC (BioLegend), and TNFα-Pacific Blue (BioLegend). Cells were acquired with an LSR-II flow cytometer (BD Biosciences). Analyses of the data were performed with the FlowJo software version 10.4.2 (Tree Star).

## Cells

B16 F10 melanoma cell line was obtained from American Type Culture Collection (ATCC, Manassas, VA, USA) and maintained in high glucose Dulbecco's Modified Eagle's Medium (Gibco-Life Technologies) supplemented with 10% FBS. Cell cultures were maintained at 37°C in a humidified incubator containing 5% $CO_2$.

## Antitumor experiment

At day 28 after mouse immunization, B16 F10 cells ($4 \times 10^5$) were s.c. injected into the mid-right flank of C57BL/6J recipient mice. Tumor growth was measured every 2–3 days with a dial caliper, and the volume was determined by ½(length×width²).

## Statistical procedures

One-way ANOVA with Dunnett correction for multiple testing was used for ELISpot analysis to establish the differences within the different groups. We analyzed the ICS results by building a bootstrap distribution based on the ratio of the percentage of specific cell types between the vaccinated and unvaccinated groups. Subsequently, we determined the centered 95% confidence interval. For statistical analysis of the overall survival in the antitumor experiment, log-rank (Mantel-Cox) statistical test was performed on day 24. The statistical significances are indicated as follows: *, $p < 0.05$; **, $p < 0.005$; ***, $p < 0.001$. Statistical analysis was performed using Prism Software.

## Ethical statement

Female C57BL/6J mice (6–8 weeks of age) used for in vivo experiments were purchased from The Jackson Laboratory and stored in the animal facility of Centro Nacional de Biotecnología (CNB-CSIC, Madrid). In vivo studies were approved by the Ethical Committee of Animal Experimentation (CEEA) of CNB-CSIC and by the competent authority of *Comunidad de Madrid* (PROEX 041.4/21). Animal procedures were conformed to international guidelines and to Spanish-European law.

## Acknowledgements

The research is supported by grants: PID2020-116393RB-I00, financed by MCIN/ AEI /10.13039/501100011033/and BFERO2020.04, financed by FERO foundation. AMP is supported by FPU18/03199JMGG from MCIN. We are grateful to the Proteomics and the Animal Facilities of CNB for peptide synthesis and their support.

## Additional information

### Funding

| Funder | Grant reference number | Author |
|---|---|---|
| Ministerio de Ciencia e Innovación | PID2020-116393RB-I00 | Esteban Veiga |
| Agencia Estatal de Investigación | 10.13039/501100011033 | Esteban Veiga |
| Fundación Fero | BFERO2020.04 | Esteban Veiga |
| Ministerio de Ciencia e Innovación | FPU18/03199JMGG | Almudena Mendez-Perez |

The funders had no role in study design, data collection and interpretation, or the decision to submit the work for publication.

### Author contributions

Almudena Mendez-Perez, Formal analysis, Investigation, Writing - review and editing; Andres M Acosta-Moreno, Investigation, Methodology; Carlos Wert-Carvajal, Conceptualization; Pilar Ballesteros-Cuartero, Data curation, Methodology; Ruben Sánchez-García, Conceptualization,

Methodology; Jose R Macias, Rebeca Sanz-Pamplona, Methodology; Ramon Alemany, Methodology, Writing - review and editing; Carlos Oscar Sorzano, Arrate Munoz-Barrutia, Conceptualization, Data curation, Formal analysis, Methodology, Writing - review and editing; Esteban Veiga, Conceptualization, Resources, Data curation, Formal analysis, Supervision, Funding acquisition, Validation, Investigation, Visualization, Methodology, Writing - original draft, Project administration, Writing - review and editing

### Author ORCIDs
Arrate Munoz-Barrutia ⓘ https://orcid.org/0000-0002-1573-1661
Esteban Veiga ⓘ https://orcid.org/0000-0002-7333-2466

### Ethics
In vivo studies were approved by the Ethical Committee of Animal Experimentation (CEEA) of CNB-CSIC and by the competent authority of Comunidad de Madrid (PROEX 041.4/21). Animal procedures were conformed to international guidelines and to Spanish-European law.

Reviewer #1 (Public review): https://doi.org/10.7554/eLife.95010.3.sa1
Author response https://doi.org/10.7554/eLife.95010.3.sa2

## Additional files

### Supplementary files
MDAR checklist

### Data availability
All data generated or analysed during this study are included in the manuscript and supporting files; source data files have been provided for Figures 1, 2 and 3.

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
