## [Editor Report · eLife Assessment]

Veiga et al demonstrate the importance of incorporating RNAseq and machine learning approaches for neoantigen prediction. The evidence is **convincing**, and these findings contribute **important** information towards the selection of neoantigens for personalized antitumor vaccination.

---

## [Referee Report · Reviewer #1 (Public review)]

Summary:

The authors of the study are trying to show that RNAseq can be used for neoantigen prediction and the machine learning approach to the prediction can reveal very useful information for the selection of neoantigens for personalized antitumor vaccination.

Strengths:

The authors demonstrated that RNA expression of a neoantigen is very important factor in the selection of peptides for the creation of personalized vaccines. They proved in vivo that in silico-predicted neoantigens can trigger antitumor response in mice.

Weaknesses:

The authors replied to my previous comment about the selection of the peptides for vaccination in the responses to reviewers, but didn't include that in the revised manuscript. I think all that information should be in the manuscript.

Here is the original comment: "The selection of the peptides for vaccination is not clear. Some peptides were selected before and some after processing. What processing is also not clear. The authors didn't provide the full list of peptides before and after processing, please add those. And it wasn't clear that these peptides were previously published. Looking at the previously published table with peptide from B16 F10 (https://www.nature.com/articles/s41598-021-89927-5/tables/3), there are other genes with high expression, e.g. Tab2, Tm9sf3 that have higher expression than Herc6, please clarify the choice."

---

## [Author Response]

The following is the authors’ response to the original reviews.

**Reply to Reviewer #1 (Public Review):**

The post-processing increases number of putative neoantigens. As shown in Author response image 1, this is done through data augmentation or “mutations” of individual amino acids in a sequence by their most similar amino acid in the BLOSUM62 embedding. If most of the mutations result in a positive prediction (which we binarize through a >0.5 score) the sequence changes its prediction.

**Author response image 1. sa2fig1:** Post-processing pipeline to increase the number of putative neoantigens. Sequences can either be predicted using the forward method, for which a raw score is produced, or it can be introduced to a majority-vote prediction of the ensemble prediction of similar protein sequences.

In this article, we obtain the following candidates after post-processing.

**Author response table 1. sa2table1:** Sequence symbol gene prediction fragments per kilobase million (FPKM).

PPTARMMFSGLA Wiz	ENSMUSG000000024050 1	16,7001
NKVVMEYENLEK Pnp	ENSMUSG00000115338 1	3,03912
TFVYPTIFPLRE Lrrc28	ENSMUSG00000030556 1	0,941842
SQAWTHPPGVVN Adar	ENSMUSG00000027951 1	0
KASGFRYNVLSC Nr1h2	ENSMUSG00000060601 1	0
FVPGPSFWGLVN Car11	ENSMUSG00000003273 1	0

As mentioned, the prediction column shows a binary label. The full list contained 402 sequences did not include any other sequences that met the majority vote criteria.

As noted by the reviewer, the Table 3 of our original paper includes the scores of the direct prediction, which has four sequences in common with the post-processing criteria (*Pnp, *Adar, *Lrrc28 and *Nr1h2). * indicates the mutated form of the peptide, i.e neoantigen.

We selected the top 4 predicted antigens present both by direct prediction and after post-processing; (*Pnp, *Adar, *Lrrc28 and *Nr1h2) (Wert-Carvajal et al. 2021), but we encountered difficulty in synthesizing, *Nr1h2 (Mutated Nr1h2), and thus it could not be included in the study.

We also decided to evaluate the immunogenicity of *Wiz, which was identified as a potential TNA only after postprocessing. *Wiz exhibited lower levels of immunogenicity compared to *Pnp, *Adar, and *Lrrc28. However, unlike these, *Wiz is highly expressed in the tumor, and vaccination with *Wiz provided the strongest protection levels. These findings led us to incorporate post-processingg into the NAPCNB platform.

We chose *Herc6 as a mutated antigen predicted not to be a TNA over other candidates because its expression in the tumor was similar to that of *Wiz.

Depending on the experiment we used 4 or 5 animals per group (this is now clarified in the revised version).

The software used for statistical analysis was GraphPad Prism.

**Reply to Reviewer #2 (Public Review):**

This is true, binding affinity does not always predict immune responses but in most cases, high affinity peptides are immunogenic. There are of course other parameters that drive the effective priming of tumor-reactive CD8+ T cells through antigen crosspresentation, but the mechanisms of antigen presentation are yet not completely understood. High affinity peptides are desirable as good candidates in neoantigen-based vaccines.

Other comments of the reviewers

**Reviewer #1 (Recommendations For The Authors):**
- Please decipher all abbreviations when they appear for the first time, e.g. NAP-CNB, PBS, CFA, FIA, and so on.

Done in the revised version.

- Please be consistent with the capitalization of gene names (WIZ vs Wiz, TRP2 vs Trp2, and so on), and why there is an asterisk.

Done in the revised version.

- Please be clear about where you use cell lines or mice as a model. It's not clear.

All work is done in mice, or cells isolated from vaccinated mice.

- Why there is an asterisk in front of gene names?

Explained in the revised version; The * indicates the peptides that are the mutated version.

- Please add a reference for the following statement in the Introduction: "However, the response rates of these therapies remain low and relapses are common."

Done in the revised version.

- Also please add a reference for the use of TRP2 as a positive control.

Done in the revised version.

**Reviewer #2 (Recommendations For The Authors):**
- It may be helpful to validate a larger pool of antigens. This is not necessary however and could be done in a follow-up study.

We are doing it for other studies with excellent results.

- The negative PBS control should be included in Figure 1.

Done in the modified figure 1C in the revised version.

- Stats should be clearly indicated in Figure 2.

Done in the revised version.

- Some nuances should be discussed. Is a threshold of neoantigen expression required or is there a correlation with tumor control? On the flip side, these neoantigens that are not likely to elicit immune responses but are highly expressed are also not likely to mediate tumor control.

These points have been discussed. Based on our data, strategies for designing antitumor therapies should prioritize antigens that are highly expressed in tumors, even if they are not the most immunogenic. However, it is worth noting that even low-expressed antigens can still elicit an antitumor immune response. If possible one should define strategies attacking multiple antigens in order to minimize tumor scape. Whenever possible, strategies should be developed to target multiple antigens simultaneously, aiming to minimize tumor escape.

- This study focuses on CD8 T cell responses but CD4s are also important in tumor control. This could be mentioned in the discussion.

This is true, but this article focuses on validating a platform that predicts the antigenicity of antigens presented in the context of MHC-I.

- Ideally, we would want to see that these responses are not elicited with adjuvant alone as an additional control.

The non-vaccinated control animals received PBS and adjuvant. This clarification has now been included in the text.